# Could Giant Pretrained Image Models Extract Universal Representations?

**Yutong Lin**[*13], **Ze Liu**[*23], **Zheng Zhang**[3], **Han Hu**[3], **Nanning Zheng**[1], **Stephen Lin**[3], **Yue Cao**[3]

[1]Xi'an Jiaotong University
[2]University of Science and Technology of China
[3]Microsoft Research Asia
{t-yutonglin,t-liuze,zhez,hanhu,stevelin}@microsoft.com
nnzheng@mail.xjtu.edu.cn,caoyue10@gmail.com

## Abstract

Frozen pretrained models have become a viable alternative to the pretraining-then-finetuning paradigm for transfer learning. However, with frozen models there are relatively few parameters available for adapting to downstream tasks, which is problematic in computer vision where tasks vary significantly in input/output format and the type of information that is of value. In this paper, we present a study of frozen pretrained models when applied to diverse and representative computer vision tasks, including object detection, semantic segmentation and video action recognition. From this empirical analysis, our work answers the questions of what pretraining task fits best with this frozen setting, how to make the frozen setting more flexible to various downstream tasks, and the effect of larger model sizes. We additionally examine the upper bound of performance using a giant frozen pretrained model with 3 billion parameters (SwinV2-G) and find that it reaches competitive performance on a varied set of major benchmarks with only one shared frozen base network: 60.0 box mAP and 52.2 mask mAP on COCO object detection test-dev, 57.6 val mIoU on ADE20K semantic segmentation, and 81.7 top-1 accuracy on Kinetics-400 action recognition. With this work, we hope to bring greater attention to this promising path of freezing pretrained image models.

## 1 Introduction

Transfer learning via the pretraining-then-finetuning paradigm is the cornerstone in the success of deep neural networks. In computer vision, finetuning backbone networks [29, 21, 13, 38] pretrained on supervised classification leads to top performance for a wide range of visual recognition tasks, even on tasks where the input or output format differs from that of pretraining, such as video action recognition [25], object detection [34], and semantic segmentation [58]. In natural language processing (NLP), finetuned models that are pretrained on masked language modeling (MLM) also excel at a large variety of NLP tasks, e.g., on eleven tasks in the case of BERT [11].

Though highly effective, the changes to many network parameters make finetuning parameter-inefficient. An entirely different model is created for each downstream task, a problem that is magnified with the rapid increase in model sizes. In NLP, a solution to this problem that has gained momentum is to freeze pretrained language models when transferring them to downstream tasks. A small number of task-specific parameters, e.g., task-specific heads [22] or prompts [3, 30, 32, 36], are introduced, and only these newly added parameters are trained on the downstream task while the

---

[*]Equal contribution. The work is done when Yutong Lin and Ze Liu are interns at Microsoft Research Asia. Correspondence to: Yue Cao (caoyue10@gmail.com).

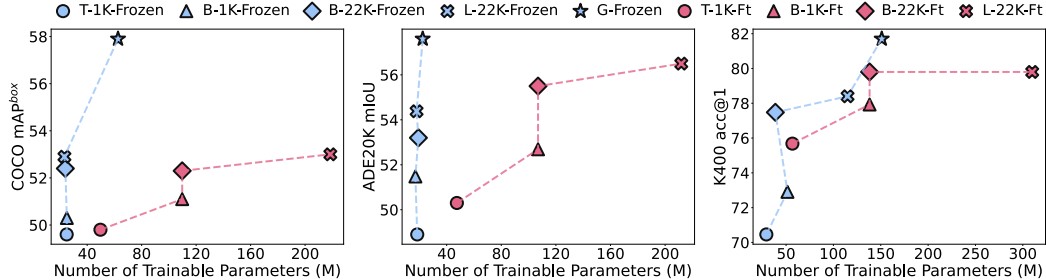

Figure 1: Performance with respect to the number of trainable parameters with different base networks (SwinV2-T/B with supervised training on ImageNet-1K, SwinV2-B/L with supervised training on ImageNet-22k, and SwinV2-G) under frozen and full finetuning settings on COCO object detection (left), ADE20K semantic segmentation (middle), and Kinetics-400 video action recognition (right). Single-scale box mAP for COCO, multi-scale mIoU for ADE20K and top-1 accuracy for Kinetics-400 are reported.

rest of the model is fixed. In this way, the computation and memory costs are reduced considerably, with a high degree of parameter sharing among models finetuned for different tasks.

This parameter-efficient transfer, however, has yet to gain traction in computer vision. The difficulties in adopting this approach lie in the differences between the two modalities. In NLP, the input and output formats of different tasks are similar, where nearly all of them can be defined as a sequence of tokens. Moreover, the dominant pretraining task of self-supervised masked language modeling learns information that is broadly valuable in NLP applications. By contrast, such uniformity does not exist in computer vision. The input and output formats in vision vary greatly, such as image/video and high/low resolution for inputs, and image-level categories, target coordinates and so on for outputs. The information that is valuable varies as well among downstream tasks, making the choice of pretraining task unclear. Due to these challenges, previous investigations have focused only on downstream tasks similar to the pretraining task, image classification [2, 42, 43, 57, 56, 59].

In this paper, we study parameter-efficient transfer learning that generalizes to a diverse set of computer vision tasks. Under this setting, our work seeks to answer the following questions:

1. Which pretraining task fits best with this frozen setting?
2. What is the key to making the frozen setting work well when the downstream tasks are significantly different from the pretraining task?
3. How well can a frozen setting perform with a giant pretrained model, such as SwinV2-G with 3B parameters?

Our investigation begins with the first question, where we consider the four most widely-studied pretraining tasks: supervised classification, contrastive learning, masked image modeling and image-text alignment. With SwinV2-B [38, 37] as the backbone, the pretraining tasks are evaluated on popular vision frameworks covering a range of downstream tasks, including Mask R-CNN [20] with FPN [33] for COCO object detection, Mask2former [9] for ADE20K semantic segmentation, and Video Swin Transformer [39] for Kinetics-400 (K400) video action recognition. We find that supervised pretraining and image-text alignment work best under the frozen setting and the model with ImageNet-22K pretraining performs better than the ImageNet-1K counterpart. However, the frozen settings still underperform full finetuning by substantial margins.

Towards improving performance over disparate tasks, we next consider how to extend the frozen setting for better task adaptation. We examine the effects of adding tunable parameters to task-specific heads and task-based architectural elements such as FPN for object detection and decoders for semantic segmentation. From a performance analysis and an examination of feature similarity across different layers by Centered Kernel Alignment (CKA) [27], we make a number of observations for different tasks on what can help to bridge the performance gap between frozen settings and full finetuning. Overall, we find that the number of well-placed tunable parameters is the key to make frozen settings work well.

Finally, we look at the impact of model size on the frozen setting. With its richer content, larger pretrained models have been found in NLP to require fewer tunable layers. Despite a large difference

between pretraining and finetuning tasks as well as a relatively small change in model size, we have observed similar trends for computer vision between SwinV2-L-22K and SwinV2-T-1K in object detection. We additionally explore the upper bound of performance under the frozen setting, using a frozen supervised pretrained model with 3 billion parameters trained on 70M images (SwinV2-G). With this giant model, highly competitive performance is achieved on major benchmarks: 60.0 box AP on COCO object detection test-dev, 57.6 mIoU on ADE20K semantic segmentation, and 81.7 top-1 accuracy on Kinetics-400 action recognition.

Thanks to the efficiency of this frozen setting, competitive performance are achieved on different model sizes with much less trainable parameters, as indicated in Figure 1. We hope our work will inspire further research in this promising direction of freezing pretrained image models. The proposed approach can serve as a simple baseline and guide the evaluation of future work.

## 2    Related Work

**Representative Visual Pretraining Tasks**    Throughout the deep learning era, supervised learning, especially by image classification on ImageNet [10], has been prevalent in vision pretraining. Leveraging a large amount of data, a supervised pretrained model can efficiently transfer to various downstream tasks [12, 28, 26, 13, 46, 16, 38, 40, 48, 6] by finetuning. However, supervised pretraining suffers from severe data-hunger, while the cost of acquiring labeled data is expensive. To address this issue, several recent studies in self-supervised pretraining have demonstrated promising results, where self-supervised pretraining achieves finetuning performance on par with the supervised counterparts on several representative downstream tasks [19, 8]. Among them, contrastive learning [14, 19, 8, 5, 17] and masked image modeling [7, 1, 18, 54] have been particularly successful. Contrastive learning compares two image views, maximizing the similarity of positive pairs while minimizing the similarity of negative pairs. For linear probing, the state-of-the-art contrastive model even achieves results comparable to supervised models. Differently, masked image modeling learns by randomly masking some input tokens and reconstructing the raw signals. With masked image modeling, a large-scale model can be trained more effectively and yields better finetuning performance. Another branches of work utilize image-text pairs and leverage natural language as supervision for visual-linguistic learning. Among them, image-text alignment pre-training [41, 24] with web-crawled data have shown great potential for visual recognition tasks, especially under the zero-shot setting. iCar [53] further bridges image-text alignment and image classification and successfully keep the characters of both methods.

These pretraining methods are so successful that almost all the top models of various vision tasks are finetuned from them. However, in the frozen backbone setting, the performance of these pretraining tasks for various downstream tasks are still unknown. Based on this motivation, we carried out a comparison study under the frozen setting with the four most widely-used pretraining tasks – supervised classification [38, 37], contrastive learning [31], masked image modeling [54] and image-text alignment [53] – using Swin Transformer as the backbone.

**Frozen Language Models**    With the large scale of today's language models, the setting of frozen language models has become important in NLP and receives much attention [35]. Adding external adapters [22] is a direct solution in this direction and was first introduced in this field. Subsequently, various solutions specific to NLP have been proposed, including prompt tuning [30], prefix tuning [32], and low-rank adaptation [23].

**Frozen Setting in Vision**    In computer vision, additive models are the most studied direction for the frozen setting, where the pretrained weights are frozen and a small number of new parameters is added for each task. Some of these works [12, 47] directly add the new layers upon off-the-shelf features, while others introduce a new network with independent access to the input [44, 57] to address the information loss of the pretrained model, or add task-specific components into the pretrained model [42, 2, 43], such as batch normalization, residual adapter and so on. However, all previous works [2, 42, 43, 57] use the additive models only for tasks similar to the image classification pretraining task. In concurrent work [52], image classification features are reused only for object detection and instance segmentation. Our paper, for the first time, studies the frozen setting under different pretrained image models on three representative but diverse vision tasks, namely object detection, semantic segmentation and video action recognition. Furthermore, we demonstrate the

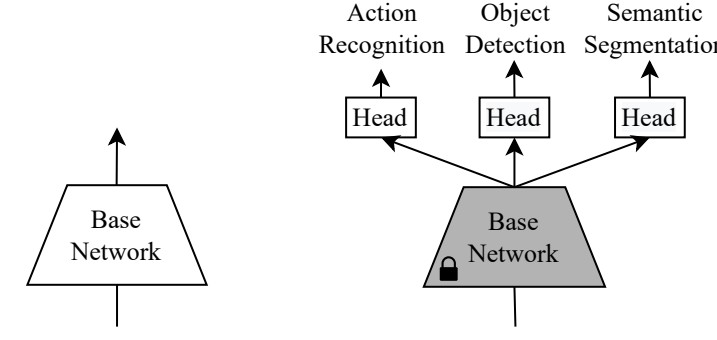

i. Pretraining of Base Network     ii. Finetuning the Head Networks

Figure 2: Pipeline of the frozen setting, consisting of (i) pretraining of the base network, and (ii) finetuning of head networks with the frozen base network.

effectiveness of our approach via a giant frozen supervised pretrained model (SwinV2-G), with highly competitive performance on major benchmarks for the three tasks. Another path is to directly adapt the prompt learning approach in NLP to computer vision [59, 56], but this is limited to very specific tasks or scenarios, e.g., requiring a text encoder [59].

## 3 Methodology

Under the frozen setting, we take the approach of freezing the pretrained base network B($\cdot$) and training a specially designed task-specific head network H($\cdot$) for each downstream task.

### 3.1 Architectural Elements

**Base Network** We denote deep neural networks designed for image classification, e.g., VGG [49], ResNet [21] or ViT [13], as the *base network*, B($\cdot$). The base network is also referred to as the backbone network in some scenarios, and it is intended to provide some core cognitive or perceptual information about the input. In the frozen setting, after the base network B($\cdot$) is pretrained, we do not update it when it is transferred to downstream tasks, that is, the base network has no tunable parameters for training on downstream tasks.

For the base network, we adopt a general-purpose backbone architecture, e.g., Swin Transformer [38] for most experiments, that is compatible with a broad range of vision tasks, without components specific to any downstream tasks. We note that Swin Transformer is highly scalable, with a giant version composed of 3 billion parameters [37].

**Head Network** We denote the task-specific network that is newly added and tuned during transfer to downstream tasks as the *head network*, H($\cdot$). Note that the input and output formats in vision vary greatly, so a crucial component of our approach is to effectively adapt the head network to the properties of the downstream task. With a suitable head design, we find this simple framework to generally perform well over the sampled downstream tasks under the frozen pretrained image model setting.

### 3.2 Pretraining

We consider the most widely-used pretraining tasks – supervised classification [38, 37], contrastive learning, masked image modeling and visual-linguistic learning – with Swin Transformer as the base network. Specifically, we utilize Swin Transformer variants specially designed for each of these pretraining tasks, e.g., vanilla Swin Transformer [38, 37] for supervised classification, EsViT [31] for contrastive learning, SimMIM [54] for masked image modeling and iCar[53] for visual-linguistic learning, which are all open-source. For fair comparison, the same base network capacity is maintained across the pretraining tasks, such as using Swin Transformer with 224$\times$224 input and a window size of 7.

### 3.3 Frozen Setting on Downstream Tasks

A major challenge of the frozen setting is the varied input and output formats of different vision tasks, such as object detection with high-resolution images as input and target coordinates and the corresponding categories as output, semantic segmentation with high-resolution images for input and pixel-level categories as output, and video action recognition with low-resolution videos for input and video-level categories as output. Therefore, adapting a frozen base network to a new task requires careful model design, including a suitable framework and head network, to bridge the gap between pretraining and finetuning.

**Object Detection** To adapt a frozen pretrained model to the object detection task, we present a few adjustments. First, object detection generally requires high-resolution images as input and a large window size in the backbone architecture, while the pretraining task takes low-resolution images and a small window size. We find that directly changing the size of input images from small to large but keeping the window size unchanged works relatively well. Second, maintaining the aspect ratio of the input image is preferable in object detection and thus by default an arbitrary input resolution is allowed for each image, with undesirable padding on the border of feature maps that change the distributions of features. We deal with this issue by adopting a multi-scale augmentation similar to that of image classification. It randomly resizes the original image, then randomly crops a square part of the resized image [15]. Third, object detection typically utilizes multi-resolution feature maps as the input of the head network, while most pretraining tasks only take the final output of the base network. Using Swin Transformer as the base network enables us to directly take the output features of different stages as the input to the head network, and we find that this choice empirically works well. For evaluation on this task, we adopt COCO 2017 [34], the most widely-used benchmark for object detection and instance segmentation, which contains 118K training, 5K validation and 20K test-dev images.

**Semantic Segmentation** Semantic segmentation aims to perform pixel-level classification on high resolution images and has many characteristics similar with object detection. For instance, semantic segmentation also requires a larger input resolution and window size, and multi-resolution feature maps for visual recognition at a finer granularity. On these two points, we obtain similar observations for semantic segmentation as with object detection: larger input with unchanged window size achieves relatively good performance, and directly using the output features from different stages of Swin Transformer as the input of the head network also works well for semantic segmentation. For this task, we adopt the most widely-used benchmark, ADE20K [58], for evaluation. ADE20K covers a broad range of 150 semantic categories. It has 25K images in total, with 20K for training, 2K for validation, and another 3K for testing.

**Video Action Recognition** The main challenge of adapting the frozen pretrained image model to video action recognition lies in the different input formats. Video action recognition aims to recognize the action types of each input video, which consists of a sequence of video frames (e.g., 16 frames per clip). Therefore, it is essential in video action recognition to capture both spatial and temporal relationships. Previous works based on full finetuning mainly focus on exploring the simultaneous modeling of spatial and temporal relationships in the base network. For the frozen setting, however, the pretraining model is capable only of spatial reasoning, so the head network must compensate for the lack of temporal modeling. For the task of human action recognition, we adopt the widely-used Kinetics-400 [25] dataset, consisting of ∼240k training videos and 20k validation videos over 400 human action categories.

## 4 Which Pretraining Task Fits Best with the Frozen Setting?

In this section, we evaluate four prevalent pretraining tasks under the frozen setting, namely supervised pretraining, contrastive learning, masked image modeling, and visual-linguistic learning. Specifically, we use six pretrained Swin Transformer models, which include supervised pretraining on ImageNet-1K (SUP-1K), supervised pretraining on ImageNet-22K (SUP-22K), contrastive learning of EsViT on ImageNet-1K (EsViT-1K), masked image modeling of SimMIM on ImageNet-1K (SimMIM-1K), image-text alignment of iCar on Laion[45](iCar-Laion), and jointly training of image-text alignment and image classification for iCar on Laion and ImageNet-22K (iCar-Laion-22K). For the downstream

tasks, we adopt SwinV2-B [37] as the base network and three widely-used frameworks as the head networks: Mask R-CNN [20] with FPN [33] for COCO object detection, Mask2Former [9] with a one-block pixel decoder for ADE20K semantic segmentation, and a spatial-only Video-Swin-Transformer [39] variant, where a temporal window size of 1 is used, with a linear head for Kinetics-400 action recognition.

| Approach | COCO | | | | ADE20K | | Kinetics-400 | |
| | Frozen | | Full ft. | | Frozen | Full ft. | Frozen | Full ft. |
| | $AP^{box}$ | $AP^{mask}$ | $AP^{box}$ | $AP^{mask}$ | mIoU | mIoU | acc@1 | acc@1 |
|---|---|---|---|---|---|---|---|---|
| SUP-1K | 42.4 | 38.7 | 50.5 | 44.5 | 49.8 | 52.3 | 60.4 | 77.0 |
| SUP-22K | 45.0 | 41.1 | 51.9 | 45.7 | 51.9 | 55.3 | 70.3 | 79.7 |
| EsViT-1K | 42.0 | 38.5 | 51.5 | 45.6 | 49.7 | 52.1 | 62.0 | 76.5 |
| SimMIM-1K | 34.1 | 32.4 | 52.9 | 46.7 | 42.4 | 51.7 | 14.2 | 75.9 |
| iCAR-Laion | 43.3 | 39.5 | 51.7 | 45.5 | 51.0 | 55.3 | 65.1 | 79.5 |
| iCAR-Laion-22K | 44.9 | 41.2 | 52.3 | 46.1 | 51.1 | 55.4 | 69.4 | 80.2 |

Table 1: Comparison of different pretraining tasks on frozen and full finetuning settings. Results of box mAP and mask mAP for COCO object detection, mIoU for ADE20K semantic segmentation, and top-1 accuracy for Kinetics-400 action recognition are reported. SUP denotes supervised classification as pretraining.

Results are shown in Table 1. For the models pretrained on the ImageNet-1K dataset, we can observe that SUP-1K and EsViT-1K perform similarly on almost all the benchmarks, including both the full finetuning and frozen settings. This phenomenon is understandable, because the linear evaluation result of EsViT is relatively high, indicating that the features extracted by EsViT-1K are similar to the SUP-1K counterpart. The SimMIM pretrained model performs competitively high in the full finetuning setting. For example, it achieves much better performance on COCO, comparable performance on ADE, and slightly worse performance on K400, in comparison to the SUP-1K counterpart. However, the SimMIM-1K model performs poorly on all benchmarks in the frozen setting. This could be foreseen from its poor linear evaluation performance on ImageNet-1K, indicating that its output features do not capture high-level semantics. The iCAR-Laion model outperforms SUP-1K and EsViT-1K on almost all the benchmarks, indicating the effectiveness of learning from large scale image-text datasets. In addition, with jointly training on Laion and ImageNet-22K, the iCAR-Laion-22K model achieves similar results with the SUP-22K model. We can also observe that the SUP-22K model outperforms SUP-1K significantly in all settings, reflecting the great benefit that data scaling brings to supervised pretraining. In general, we find that supervised pretraining works best under the frozen setting, and we adopt supervised pretrained models by default in the following experiments. However, in all three downstream tasks, the frozen settings still underperform full finetuning by substantial margins.

## 5 What is the Key to Making the Frozen Setting Work?

Towards improving performance over disparate tasks, in this section we consider how to extend the frozen setting for better task adaptation. For the following investigation, we utilize the best-performing pretraining task, supervised classification on ImageNet-22K (SUP-22K).

### 5.1 Adding More Tunable Parameters at Head Networks

To some extent, the poor performance of the frozen setting is understandable. Despite the significant difference between the pretraining and downstream task, all parameters in the base networks are locked in the frozen setting, and only a few newly-added parameters are available for task adaptation. Thus, a direct solution is to add more tunable parameters to the task-specific head networks. As the head network varies among different downstream tasks, we present task-specific strategies on how to add parameters to the head.

**Object Detection**    For object detection, we adopt the framework of Mask R-CNN [20] with FPN [33] as the head network. Previous work on Mask R-CNN mainly explored improvements over FPN and the box/mask head, so we also examine common alternatives to these two components, e.g., replacing

| Head Network | Frozen | | Full ft. | |
|---|---|---|---|---|
| | $AP^{box}$ | $AP^{mask}$ | $AP^{box}$ | $AP^{mask}$ |
| FPN | 45.0 | 41.1 | 51.9 | 45.7 |
| FPN w. $4\times$ Residual Blocks inside | 49.5 | 44.0 | 52.1 | 45.6 |
| BiFPN | 51.9 | 46.0 | 52.3 | 45.7 |
| FPN w. Cascade Head | 49.0 | 43.0 | 54.5 | 46.9 |
| BiFPN w. Cascade Head | 53.8 | 46.7 | 54.3 | 46.9 |

Table 2: Results for different head networks on COCO object detection and instance segmentation, including FPN with vanilla box/mask head, FPN with four more residual blocks inside, BiFPN, FPN with cascade head, and BiFPN with cascade head.

FPN with BiFPN [51] or adding more residual blocks in FPN, and changing the vanilla box/mask head to a cascaded box/mask head [4]. Results are shown in Table 2. We can observe that adding parameters to FPN, e.g., replacing FPN with BiFPN or adding more residual blocks, both significantly bridge the gap between the frozen setting and full finetuning, from -6.9 to -2.6 box mAP and -6.9 to -0.4 box mAP, respectively. As BiFPN is carefully designed with many interactions between multi-resolution feature maps, replacing FPN with BiFPN outperforms the model of adding more residual blocks in FPN. However, replacing the box/mask head with a cascade head [4] improves the performance of both the frozen and full finetuning settings, from 45.0 to 49.0 box mAP and 51.9 to 54.5 box mAP, but does not bridge the gap between them. Adding more tunable parameters to FPN is thus found to be more effective than adding to the box/mask head.

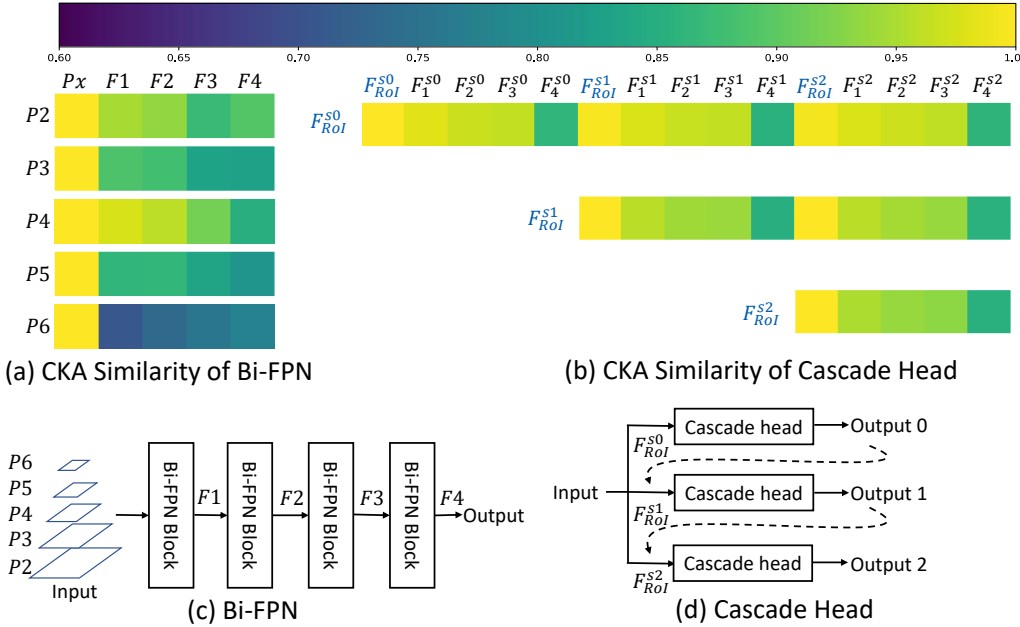

Figure 3: The CKA heatmap on the features across layers in BiFPN (a) and Cascade Head (b). The detailed architecture of BiFPN (c) and Cascade Head (d).

To understand this difference in behavior, we conduct a performance analysis and an examination of feature similarity across different layers in BiFPN and Cascade Head by Centered Kernel Alignment (CKA) [27]. As shown in Figure 3 (a), for each stage (with different resolutions of feature map) of BiFPN, we plot the CKA similarity between the input features ($Px$ of stage $x$) and the output of each block ($Fx$ of block $x$). For the cascade head, as shown in Figure 3 (b), we plot the CKA similarity between the input features of each stage ($F_{RoI}^{si}$ in stage $i$) and the hidden features inside each stage ($F_j^{si}$: output of $j$'s block in stage $i$). From this figure, we can observe that the features across layers and even stages in cascade heads are almost the same except for the last output, but in BiFPN, features across different layers are different (with lower CKA similarity). In other words, adding parameters in BiFPN provide more useful capacity for transformation, but adding parameters

in the cascade head hardly provides useful computation. This difference can be explained by the architecture of BiFPN (Figure 3(c)) and cascade head (Figure 3(d)). BiFPN follows a sequential structure, where the input of each block is the output of the previous block. The cascade head follows a parallel structure, and all stages extract RoI features from the original multi-resolution feature map, making the input of each stage have extremely high CKA similarity. Also, the transformations made by each stage are similar. The last output of the cascade head behaves differently because it is followed by a pooling layer, which is proven to have a great impact on CKA similarity. Consequently, the number of well-placed tunable parameters is the key to make frozen settings work well.

| Head Network | Frozen mIoU | Full Ft. mIoU |
|---|---|---|
| 1× Pixel Dec. w. (3+1)× Transformer Dec. | 51.0 | 55.4 |
| 6× Pixel Dec. w. (3+1)× Transformer Dec. | 52.6 | 55.3 |
| 1× Pixel Dec. w. (9+1)× Transformer Dec. | 51.9 | 55.3 |
| 6× Pixel Dec. w. (9+1)× Transformer Dec. | 53.2 | 55.5 |

Table 3: Results from using different head networks on ADE20K semantic segmentation, with either one or six pixel decoder blocks, and a Transformer decoder with four or ten blocks.

**Semantic Segmentation** For semantic segmentation, we adopt Mask2former [9] with a one-block pixel decoder and four-block transformer decoder as the head network. The framework of Mask2former is similar to Mask R-CNN in object detection, where the pixel decoder corresponds to the FPN, and the transformer head corresponds to the box/mask head. Thus, we employ a setting similar to that used for object detection, by increasing the number of parameters in these two major components, e.g., changing the pixel decoder to six blocks and the transformer decoder to ten blocks. As shown in Table 3, similar results can be observed where enlarging the pixel decoder helps more in bridging the gap than enlarging the transformer head. We also perform a CKA analysis on this framework and observe behaviors like that in object detection, as shown in the Appendix.

| Head Network | Frozen top-1 accuracy | Full Ft. top-1 accuracy |
|---|---|---|
| Linear | 70.3 | 79.7 |
| 4× Temporal Blocks | 74.7 | 79.5 |
| 4× Global Blocks | 77.5 | 79.8 |

Table 4: Results with different head networks on Kinetics-400 video action recognition: linear classifier, four temporal Transformer blocks, and four global (spatiotemporal) Transformer blocks.

**Video Action Recognition** For video action recognition, it is crucial to capture both spatial and temporal relationships, but the pretraining model is only capable of spatial reasoning. Under the frozen setting, the lack of temporal modeling thus needs to be compensated for in the head network. In our experiments, we adopt a spatial-then-temporal Video-Swin-Transformer [39] framework that is enhanced with additional Transformer blocks, either temporal-only or global (spatiotemporal). In Table 4, we can observe that under the full finetuning setting, adding global blocks and adding temporal blocks work similarly with a linear classifier. On the other hand, adding temporal blocks in the frozen setting leads to significantly better performance than the linear classifier setting, which verifies the need for temporal reasoning. Also, adding global blocks in the frozen setting improves substantially over adding temporal-only blocks. Although the base network is pretrained for spatial reasoning and global blocks are added, there is still a non-trivial gap between the frozen and full finetuning settings. This may be due to some missing information from the original features, and the remaining gap may be reduced by further introducing a network with independent access to the input, which is beyond the scope of this paper and left for future work.

## 5.2 Does Larger Pretrained Base Network Need Smaller Head Network?

When the pretrained model becomes larger, it generally contains richer and more valuable information, so for downstream tasks, we generally expect that a larger pretrained model does not require as large of a head. In NLP, it is verified that only a few layers or a few prompts are needed when the

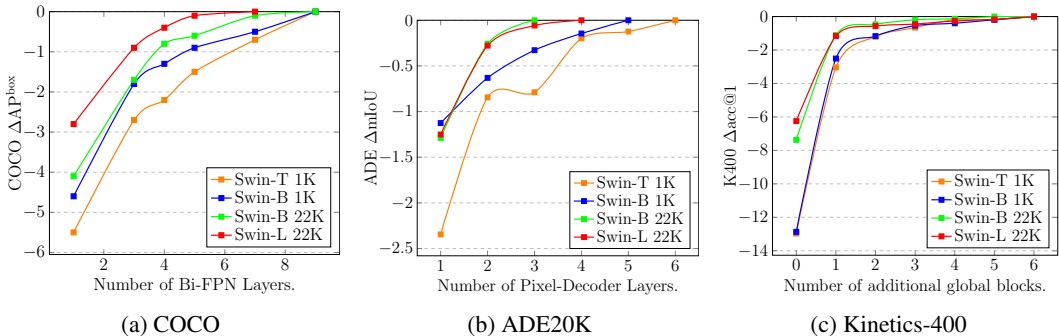

|           | (a) COCO | (b) ADE20K | (c) Kinetics-400 |

Figure 4: Performance improvements on COCO object detection, ADE semantic segmentation, and Kinetics-400 action recognition with increases in the number of trainable parameters.

pretrained model is extremely large, e.g., 173B. For computer vision, we seek an answer to whether the current best pretrained models have similar properties. We perform the empirical study with different sizes of pretrained models, SwinV2-T/B with ImageNet-1K pretraining, and SwinV2-B/L with ImageNet-22K pretraining, on the three representative downstream tasks. From Figure 4 (a), on COCO object detection, we can surprisingly observe a clear trend, that SwinV2-L-22K requires fewer parameters to converge (a 5-layer BiFPN), while SwinV2-T-1K requires more parameters to converge (a 9-layer BiFPN). On ADE20K semantic segmentation in Figure 4 (b), we also find that SwinV2-T-1K requires more parameters to converge (a 6-layer pixel decoder) than other larger base networks. On Kinetics-400 video action recognition in Figure 4 (c), different models appear to require a similar number of parameters to converge. This observation matches the previous one, that after careful tuning, there is still a non-trivial gap between the frozen and full finetuning settings on video action recognition (as shown in Table 4), indicating a larger difference between pretraining and video recognition tasks than others.

## 6 Results with Different Sizes of Base Networks

| Base Network | Head Network | COCO | | | | ADE20K | | Kinetics-400 | |
|---|---|---|---|---|---|---|---|---|---|
| | | Frozen | | Full ft. | | Frozen | Full ft. | Frozen | Full ft. |
| | | AP$^{box}$ | AP$^{mask}$ | AP$^{box}$ | AP$^{mask}$ | mIoU | mIoU | acc@1 | acc@1 |
| SwinV2-T-1K | Mask. | 49.6 | 44.0 | 49.8 | 43.8 | 48.9 | 50.3 | 70.6 | 75.7 |
| SwinV2-B-1K | Mask. | 50.3 | 44.5 | 51.1 | 44.8 | 51.5 | 52.7 | 73.3 | 77.9 |
| SwinV2-B-22K | Mask. | 52.5 | 46.6 | 52.3 | 45.7 | 53.2 | 55.5 | 77.7 | 79.8 |
| SwinV2-L-22K | Mask. | 53.0 | 46.8 | 53.0 | 46.5 | 54.4 | 56.5 | 78.7 | 79.8 |
| SwinV2-B-22K | Cascade. | 53.8 | 46.7 | 54.3 | 46.9 | - | - | - | - |
| SwinV2-G-ext22K | HTC | 57.9 | 50.4 | - | - | 57.6 | - | 81.7 | - |
| | HTC$^{\dagger}$ | 59.3 | 51.6 | - | - | | | | |

Table 5: Comparison of different-sized base networks with carefully designed head networks on frozen and full finetuning settings, including SwinV2-T/B pretrained on ImageNet-1K, SwinV2-B/L pretrained on ImageNet-22K, and SwinV2-G pretrained on ext22K. Results of box mAP and mask mAP for COCO object detection validation set, mIoU for ADE20K semantic segmentation, and top-1 accuracy for Kinetics-400 action recognition are reported. Mask. denotes Mask R-CNN is used as the head network, and Cascade. refers to Cascade Mask R-CNN. $^{\dagger}$ denotes multi-scale testing on COCO.

After careful design of the head networks for each downstream task, we compare different sizes of base network with the best setup of frozen and full finetuning settings, including SwinV2-T/B pretrained on ImageNet-1K, and SwinV2-B/L pretrained on ImageNet-22K, shown in Table 5. On COCO object detection, the performance gap is well bridged, where SwinV2-B/L-22K even achieve on-par performance under the frozen and full finetuning settings with two different head networks. But on ADE20K, there is still a significant performance gap. For example, using SwinV2-L-22K as the base network, the frozen setting still underperforms the full finetuning setting by 2.1 mIoU. A large performance gap between the frozen and full finetuning settings can also be observed on Kinetics-400

video action recognition. Interestingly, the performance gaps on ImageNet-1K pretrained models are significantly larger than that for models pretrained on ImageNet-22K, indicating that enlarging the pretraining dataset from ImageNet-1K to ImageNet-22K helps to bridge the task gap of pretraining and finetuning on action recognition.

We additionally explore the upper bound of performance under the frozen setting, using a frozen supervised pretrained model with 3 billion parameters trained on 70M images (SwinV2-G), shown in Table 5. With this giant model, highly competitive performance is achieved on major benchmarks: 59.3/60.0 box mAP and 51.6/52.2 mask mAP on COCO validation/test-dev sets, 57.6 mIoU on ADE20K semantic segmentation, and 81.7 top-1 accuracy on Kinetics-400 action recognition.

# 7    When Does Frozen Setting Outperform Full Finetuning?

To further explore the potential of the frozen setting, we compare it with full finetuning in the low-data scenario. More specifically, we test these two settings with a small portion of the COCO training dataset. Following the standard practice in semi-supervised learning [50, 55], we construct two datasets with only 1% and 10% (about 12K and 120K training images, respectively) of COCO.

| Head Network | 1% data | | | | 10% data | | | |
| --- | --- | --- | --- | --- | --- | --- | --- | --- |
| | Frozen | | Full ft. | | Frozen | | Full ft. | |
| | $AP^{box}$ | $AP^{mask}$ | $AP^{box}$ | $AP^{mask}$ | $AP^{box}$ | $AP^{mask}$ | $AP^{box}$ | $AP^{mask}$ |
| FPN | 27.5 | 26.4 | 27.5 | 25.8 | 39.1 | 36.2 | 42.2 | 37.5 |
| $1\times$ BiFPN Layer | 29.8 | 27.6 | 26.3 | 24.7 | 42.8 | 38.5 | 42.0 | 37.3 |
| $2\times$ BiFPN Layer | 31.4 | 28.4 | 26.3 | 24.6 | 44.2 | 39.4 | 42.1 | 37.3 |
| $3\times$ BiFPN Layer | 32.1 | 28.9 | 25.9 | 24.3 | 44.6 | 39.7 | 42.3 | 37.6 |
| $4\times$ BiFPN Layer | 31.9 | 28.6 | 26.0 | 24.1 | 45.0 | 40.0 | 42.1 | 37.4 |

Table 6: Results for different head networks on COCO with 1% and 10% labeled data. Head networks include FPN and BiFPN.

Results are shown in Table 6. For training with 1% labeled data, when adopting FPN as head, the frozen setting achieves performance competitive with the full finetuning one. With an increasing number of BiFPN layers, we can observe stable performance improvements for the frozen setting and inferior performance for full finetuning, which indicates an overfitting issue for the full finetuning setting. For training with 10% data, there is a performance gap of -3.1 box mAP between the frozen setting and full finetuning with FPN as the head network. Compared with training on the full COCO training dataset, this gap is reduced from -6.9 box mAP to -3.1 box mAP. When replacing FPN with BiFPN, the frozen setting surpasses full finetuning with a clear margin of 2.9 box AP and 2.6 mask AP.

# 8    Conclusion

In this paper, we presented a detailed empirical study of frozen pretrained models for diverse computer vision tasks. Based on our observations, we proposed strategies for extending the frozen setting to work effectively for various downstream tasks, and also found that a universal representation with solid performance over assorted benchmarks can be learned for a giant frozen model. Our study highlights the strong potential of this transfer learning approach, and we hope it will kindle further interest in this research direction.

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
