## A  Memory and Speed Analysis

We carefully analyzed the training speed and memory usage of the frozen backbone with additional tunable parameters and full finetuning on COCO object detection, ADE20K semantic segmentation and K400 video action recognition using various sized models from SwinV2-T to SwinV2-G. As shown in Table 1 and 2, even with more trainable parameters in the head network, training with a frozen backbone can significantly improve speed and reduce memory cost, especially for large-scale models. Moreover, freezing the backbone reduces the memory consumption of a billion-level model to less than 32G, which makes it possible to run on regular GPUs, thus helping institutions with limited resources to take advantage of such large models.

| Base Network | COCO | | | ADE20K | | | Kinetics-400 | | |
|---|---|---|---|---|---|---|---|---|---|
| | Batch Size | Frozen 5× BiFPN | Full ft. FPN | Batch Size | Frozen 6× Pix. Dec. | Full ft. 1× Pix. Dec. | Batch Size | Frozen 4× Blocks | Full ft. Linear |
| SwinV2-T | 16 | 9.74G | 9.78G | 16 | 4.24G | 5.04G | 64 | 10.43G | 15.87G |
| SwinV2-B | 16 | 10.38G | 17.29G | 16 | 4.42G | 7.84G | 64 | 13.46G | 31.47G |
| SwinV2-L | 16 | 10.56G | 25.81G | 16 | 4.84G | 11.77G | 32 | 12.80G | 27.05G |
| SwinV2-G | 8 | 31.05G | >80G | 8 | 23.84G | 78.77G | 16 | 30.54G | >80G |

Table 1: Memory usage for models of different sizes.

| Base Network | COCO | | | ADE20K | | | Kinetics-400 | | |
|---|---|---|---|---|---|---|---|---|---|
| | Batch Size | Frozen 5× BiFPN | Full ft. FPN | Batch Size | Frozen 6× Pix. Dec. | Full ft. 1× Pix. Dec. | Batch Size | Frozen 4× Blocks | Full ft. Linear |
| SwinV2-T | 16 | 0.44s | 0.46s | 16 | 0.28s | 0.34s | 64 | 0.31s | 0.51s |
| SwinV2-B | 16 | 0.50s | 0.65s | 16 | 0.31s | 0.41s | 64 | 0.52s | 0.96s |
| SwinV2-L | 16 | 0.57s | 0.84s | 16 | 0.34s | 0.48s | 32 | 0.45s | 0.78s |
| SwinV2-G | 8 | 1.20s | - | 8 | 1.23s | 3.08s | 16 | 1.14s | - |

Table 2: Speed of each iteration for models of different sizes.

## B  ADE20K with UPerNet vs. Mask2former

For ADE20K semantic segmentation, we further adopt UPerNet [9], another widely used framework, to conduct comparisons on different pretrained models. We first evaluate four different pretrained Swin Transformers [7] with UPerNet, including supervised pretraining on ImageNet-1K (SUP-1K), supervised pretraining on ImageNet-22K (SUP-22K), contrastive learning of EsViT[5] on ImageNet-1K (EsViT-1K), and masked image modeling of SimMIM [10] on ImageNet-1K (SimMIM-1K). Results are shown in Table 3. Similar to the results of Mask2former, we can find that the SUP-22K model works the best in both frozen and finetuning settings. The SUP-1K and the EsViT-1K models have competitive results. The SimMIM-1K model achieves performance similar to the SUP-1K model in the finetuning setting but lags behind other models in the frozen setting.

| Approach | Frozen | | Full ft. | |
|---|---|---|---|---|
| | mIoU | mIoU (ms+flip) | mIoU | mIoU (ms+flip) |
| SUP-1K | 43.9 | 45.5 | 49.3 | 50.2 |
| SUP-22K | 48.8 | 50.0 | 51.3 | 52.2 |
| EsViT-1K | 41.8 | 43.4 | 48.8 | 49.7 |
| SimMIM-1K | 26.0 | 27.6 | 48.6 | 49.3 |

Table 3: Comparisons of different pretraining tasks on frozen and full finetuning settings with SwinV2-B as the base network. An UPerNet framework is adopted. Results of mIoU for ADE20K semantic segmentation are reported. SUP denotes supervised classification as pretraining. *ms+flip* denotes multi-scale testing with horizontal flip augmentation.

Even though the SUP-22K model preforms best among different pretrained models, there is a gap of -2.5 mIoU between the frozen setting and full finetuning. We thus add more parameters in the head network and see if this could close the gap. As UPerNet has an FPN-like head network, we

add parameters by replacing FPN with BiFPN. As shown in Table 4, with a 5-layer BiFPN, the performance gap between the frozen setting and the full finetuning is reduced to -0.2 mIoU.

| Head Network | Frozen | | Full ft. | |
| --- | --- | --- | --- | --- |
| | mIoU | mIoU (ms+flip) | mIoU | mIoU (ms+flip) |
| FPN | 48.8 | 50.0 | 51.3 | 52.2 |
| $1\times$ BiFPN Layer | 49.3 | 50.5 | 51.4 | 52.2 |
| $2\times$ BiFPN Layer | 49.8 | 50.8 | 51.4 | 52.3 |
| $3\times$ BiFPN Layer | 50.8 | 51.4 | 51.5 | 52.2 |
| $4\times$ BiFPN Layer | 50.7 | 51.9 | 51.6 | 52.5 |
| $5\times$ BiFPN Layer | 51.3 | 52.6 | 51.5 | 52.4 |

Table 4: Comparisons with different head networks on ADE20K semantic segmentation: FPN or BiFPN with different layers, using UperNet as the segmentation framework and SwinV2-B with SUP-22K training as the base network. *ms+flip* denotes multi-scale testing with horizontal flip augmentation.

## C  CKA Analysis on Mask2Former

To further understand the behavior of Mask2Former on ADE20K semantic segmentation, we conduct a similar Centered Kernel Alignment (CKA) [4] analysis of the feature similarity across different layers in the pixel decoders and Transformer decoders as done for object detection on COCO. As show in Figure 1 (a), for each stage (with different resolutions of feature map) of the pixel decoder, we plot the CKA similarity between the input features ($Px$ of stage $x$) and the output of each block ($Fx$ of block $x$). For the Transformer decoder, as shown in Figure 1 (b), we plot the CKA similarity between the output features ($Fx$ of head $x$) of each Transformer decoder. From this figure, we can observe that the features across heads in the Transformer decoder are almost the same. But in the pixel decoder, features across different blocks are somewhat different (with lower CKA similarity). In other words, adding parameters in the pixel decoder provides more useful capacity for transformation, but adding parameters in the Transformer decoder provides much less useful computation. This well matches the previous observation on COCO with BiFPN and cascade head (shown in Figure 3 of the main paper).

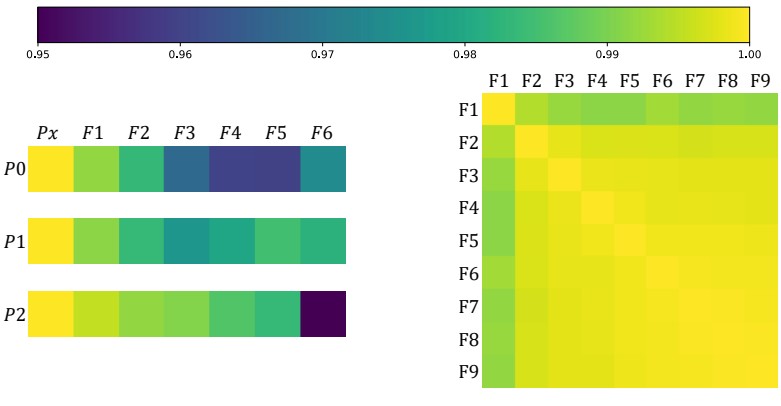

(a) CKA Similarity of Pixel-Decoder  (b) CKA Similarity of Transformer-Head

Figure 1: CKA heatmap on the features across different layers in the pixel decoder (a) and Transformer decoder (b) in Mask2former on ADE20K semantic segmentation.

## D  Detailed Settings

**Object Detection and Instance Segmentation on COCO**  Object detection and instance segmentation experiments are conducted on COCO 2017 [6]. We adopt a Mask R-CNN framework and the large-scale jittering augmentation [3]. Detailed configurations are listed in Table 5.

| Hyperparameters | 1% | | 10% | | 100% | |
|---|---|---|---|---|---|---|
| | **Frozen** | **Full Ft.** | **Frozen** | **Full Ft.** | **Frozen** | **Full Ft.** |
| Detector | Mask RCNN | | Mask RCNN | | Mask RCNN | |
| Training input size | (1024, 1024) | | (1024, 1024) | | (1024, 1024) | |
| Training scale ratio | (0.1, 2.0) | | (0.1, 2.0) | | (0.1, 2.0) | |
| Rand horizontal flip | 0.5 | | 0.5 | | 0.5 | |
| Test input size | (1333, 800) | | (1333, 800) | | (1333, 800) | |
| Training epochs | 12 | | 12 | | 72 | |
| Warm-up iterations | 500 | | 500 | | 500 | |
| Batch size | 16 | | 16 | | 16 | |
| Layer decay | ✗ | 0.95 | ✗ | 0.95 | ✗ | 0.95 |
| Base learning rate | 1e-3 | 1e-5 | 3e-4 | 3e-5 | 3e-4 | 3e-5 |
| Weight decay | 0.5 | 0.1 | 0.1 | 0.05 | 0.05 | 0.05 |
| Optimizer | AdamW | | AdamW | | AdamW | |
| Adam $\beta$ | (0.9, 0.999) | | (0.9, 0.999) | | (0.9, 0.999) | |
| Learning rate scheduler | Multi-Step | | Multi-Step | | Multi-Step | |
| Step $\gamma$ | 0.1 | | 0.1 | | 0.1 | |
| Step epochs | (8, 11) | | (8, 11) | | (63, 69) | |
| Stochastic depth | 0.3 | | 0.3 | | 0.3 | |

Table 5: Hyperparameters for the frozen setting and the full finetuning on COCO object detection dataset with 1%, 10% and all training data.

**Semantic Segmentation on ADE20K**  For the semantic segmentation task, we adopt widely-used ADE20K [11] as the benchmark. Experiments are conducted with both Mask2former [2] and UPerNet [9]. All experiments follow the settings listed in Table 6 except for those with the SimMIM-1K pretrained model. For SimMIM-1K with Mask2former, we use a learning rate of 3e-4 for full finetuning. For SimMIM-1K with UPerNet, we use a learning rate of 2e-4 and a weight decay of 0.05 for full finetuning.

| Hyperparameters | Mask2former | | UPerNet | |
|---|---|---|---|---|
| | **Frozen** | **Full Ft.** | **Frozen** | **Full Ft.** |
| Training input size | (512, 512) | | (512, 512) | |
| Training scale ratio | (0.5, 2.0) | | (0.5, 2.0) | |
| Rand horizontal flip | 0.5 | | 0.5 | |
| PhotoMetricDistortion | ✓ | | ✓ | |
| Test input size | (2048, 512) | | (2048, 512) | |
| Training iterations | 160,000 | | 160,000 | |
| Warm-up iterations | 0 | | 1,500 | |
| Batch size | 16 | | 16 | |
| Backbone lr ratio | ✗ | 0.1 | ✗ | ✗ |
| Layer decay | ✗ | ✗ | ✗ | 0.95 |
| Base learning rate | 3e-4 | 1e-4 | 2e-3 | 2e-5 |
| Weight decay | 0.02 | 0.05 | 0.05 | 0.01 |
| Optimizer | AdamW | | AdamW | |
| Adam $\beta$ | (0.9, 0.999) | | (0.9, 0.999) | |
| Learning rate scheduler | Poly (power=0.9) | | Linear | |
| Stochastic depth | 0.3 | | 0.3 | |

Table 6: Hyperparameters for the frozen setting and full finetuning on ADE20K semantic segmentation.

**Video Action Recognition on Kinetics-400**  Video action recognition experiments are evaluated on the Kinetics-400 dataset. We follow Video Swin Transformer [8] for most of the settings. Detailed hyperparameters are shown in Table 7.

**Swin-G Setting**  To explore the performance upper bound under the frozen setting, we use a strong training setting for the Swin-G model. For COCO object detection, we adopt a framework of HTC [1, 7] and a large scale jittering augmentation with an input size of (1024, 1024). The window

| Hyperparameters | Froze    Full Ft. |
|---|---|
| Training Input size | (16, 224, 224) |
| Patch size | (1, 4, 4) |
| Rand horizontal flip | 0.5 |
| Rand resized crop | ✓ |
| Training scale ratio | (0.5, 2.0) |
| Test view | 4×3 |
| Training epochs | 30 |
| Warm-up epochs | 2.5 |
| Batch size | 64 |
| Base learning rate | 3e-4 |
| Weight decay | 0.05 |
| Optimizer | AdamW |
| Adam $\beta$ | (0.9, 0.999) |
| Learning rate scheduler | Cosine |
| Stochastic depth | 0.2 |

Table 7: Hyperparameters for the frozen setting and full finetuning on Kinetics-400 video action recognition.

size is set as 32, the learning rate is 6e-4, and the batch size is 32. A weight decay of 0.05 and a stochastic depth rate of 0.3 are used. For ADE20K semantic segmentation, we set the image input size as (640, 640) and the window size as 40. The learning rate is 3e-4 and the batch size is 16. A weight decay of 0.02 and a stochastic depth rate of 0.3 are used. For Kinetics-400 video action recognition, we use a window size of 16, a learning rate of 2e-4, a stochastic depth of 0.1 and a batch size of 128.