# OpenReview forum: "Could Giant Pre-trained Image Models Extract Universal Representations?"
_NeurIPS.cc/2022/Conference — NeurIPS 2022 Accept_

### Official Review · Reviewer_Fp53 · 2022-07-07

**Rating:** 5
**Confidence:** 5
**Soundness:** 4 excellent
**Presentation:** 3 good
**Contribution:** 3 good

**Summary:**

Authors present a relatively comprehensive study of how far frozen pre-trained models can get us for multiple downstream computer vision tasks: object detection, semantic segmentation and action classification. They specifically explore 1) which pretraining task leads to most useful representations (out of supervised, self-supervised and masked image modeling, supervised on In22K is best); 2) How is the performance affected by adding more tunable parameters (it helps significantly, but less so for larger frozen models -- similar to trends observed for large language models) and 3) What is the best results one can achieve with a very large frozen model (Swin-V2-G, 3B parameters). They obtain some good results on all the 3 tasks considered.

**Questions:**

I would be interested in seeing more analysis on computational savings (i.e. training time and GPU memory) of adding additional tunable parameters on frozen backbone VS finetuning the frozen backbone. If that improves accessibility of SOTA models to users with limited resources, that would increase the significance of this work.

**Strengths And Weaknesses:**

## Strengths

1. [Significance] Impressive results with a frozen backbone: The results reported without finetuning the backbone are quite impressive and to my knowledge not previously reported. For instance 79.6% on Kinetics-400 with a frame-level feature extractor + few temporal/global aggregation blocks is comparable to recent work like TimeSformer, which is finetuned end-to-end. However it is obtained by adding additional tunable parameters on top of the frozen model, so it's not clear how favorably it compares in training compute cost to full finetuning.

2. [Clarity] The paper is well written and easy to follow. The sections are organized well and flow naturally, making the paper an easy and pleasant read.

## Weaknesses

1. [Novelty] The results aren't particularly novel: While they haven't been reported before to my knowledge and hence are interesting for the community to know, they aren't surprising or unexpected. ImageNet-22K models have been well known to generalize better to downstream task; and masked-image-modeling based models have not generalized well without full finetuning. Adding more tunable parameters is expected to improve performance.

2. [Quality] Analysis of training speed/memory usage: What does training with frozen backbone enable? One important gain would be making state of the art models accessible to users with limited resources. However the paper doesn't show explicit comparison on this axis. How does the training time of frozen + tunable parameter models compare to finetuning the frozen model without tunable parameters? Does it lead to significant training time or GPU memory savings that enable SOTA models to be trained on very small GPUs? Or do the additional trainable parameters somewhat nullify the gains by freezing the backbone?

3. [Clarity] Some parts of the model enhancements are not clear. What is the precise design of the "global" and "temporal" blocks added in Table 4? Also, how is the aggregation over the full video done?

4. [Quality] Additional analysis: While the paper has decent amount of analysis, it is limited to Swin Transformer based architectures; similar analysis on ViTs would make the paper stronger. Also, the set of tasks is somewhat limited -- object detection and semantic segmentation are quite similar (and authors also observe similar trends for both). Paper's impact can be increased by adding few more computer vision tasks, such as vision-language tasks (VQA, captioning etc), 3D/RGBD recognition tasks etc.

## Overall
The paper is generally well written and motivated. The results are interesting however not totally novel or surprising. More analysis of the computational savings would make future versions of the paper stronger. Hence, I am borderline, however leaning towards acceptance given the range of results shown across object detection, segmentation and video classification, with different sized Swin models etc.

---

> ### Author Response · Authors · 2022-08-02
> **Response to Reviewer Fp53**
>
> Thanks for the careful and valuable comments.
>
> **Q1 Novelty**
>
> We would first refer the reviewer to the **Technical Novelty** in our general response. In the general response, we discuss that this paper aims to provide a simple and effective solution for the less-studied problem of the frozen models. The studies on different pretraining tasks, data scales, model sizes and adding tunable parameters are interesting to share with the community.
>
> Please note that adding more tunable parameters is intuitive but not trivial. We provide further intuition that both the number of tunable and well-placed parameters is the key to making frozen settings work well. Moreover, we hope our work would serve as a strong start for studies on frozen models and would bring more attention to this promising topic.
>
> **Q2 Analysis of training speed/memory usage**
>
> We have listed the detailed training speed and memory usage in the **Speed/Memory analysis** of the general response. Tables in the general response show that compared to full finetuning, adding tunable parameters to the frozen backbone would significantly improve speed and reduce memory cost. This would enable users with limited resources to leverage large SOTA models.
>
> **Q3 "What is the precise design of the "global" and "temporal" blocks added in Table 4? Also, how is the aggregation over the full video done?"**
>
> "Global" indicates we compute global attention with all tokens across different frames and 2D locations, e.g., TxHxW tokens for input with shape (T, H, W).
>
> "Temporal" indicates we only compute attention on the dimension "T", e.g., T tokens of different frames at the same 2D location (x, y) for input with shape (T, H, W).
>
> For frozen models with linear heads, the aggregation is simply performed by an average pooling operation over all frames and spatial locations on top of the backbone. For frozen models with global blocks in the head network, aggregation can be achieved by global attention.
>
> **Q4 "...similar analysis on ViTs would make the paper stronger."**
>
> Thanks for this suggestion. We have added the ViT results on K400 video action recognition in the **Analysis on ViTs** of the general response. It is observed that results are similar to those using Swin.
>
> **Q5 "Paper's impact can be increased by adding few more computer vision tasks, such as vision-language tasks (VQA, captioning, etc), 3D/RGBD recognition tasks, etc."**
>
> Thanks for the suggestion. This is an interesting direction to explore, and we leave this as future work.

---

> > ### Comment · Reviewer_Fp53 · 2022-08-09
> > **Rebuttal response**
> >
> > I have looked through the authors' rebuttal. It's good to see that proposed approach leads to saving in train time/GPU memory, and also seems to generalize somewhat to ViTs. Authors should add the additional results into the final manuscript. I plan to keep my original rating.

---

> > > ### Author Response · Authors · 2022-08-09
> > > **Response to Reviewer Fp53**
> > >
> > > Thanks a lot for helping us improve this paper. We would add the additional results and discussions accordingly in the revision.

---

### Official Review · Reviewer_RULo · 2022-07-11

**Rating:** 4
**Confidence:** 4
**Soundness:** 3 good
**Presentation:** 4 excellent
**Contribution:** 2 fair

**Summary:**

This paper studies way to use large, pretrained image models for many different tasks. The approach pre-trained a swin transformer model using either supervised learning, contrastive learning or masked image modeling. This model is then frozen, and new layers are added and trained for the specific task. This paper finds that supervised pretraining transfers best and adding more layers for the finetuning is generally beneficial.

**Questions:**

Please address the weaknesses above. Specifically,

(1) Will these results change with different pretraining data scales?
(2) Are these methods really the key to using frozen image models? Are there other methods that could be better?
(3) If this paper is trying to find the key to using frozen models, why are the results lower than finetuned? Is this an inherent limitation of using frozen models? Or has the method to properly utilize pretrained models not been found?

**Limitations:**

There was no impact section and the only limitations mentioned were that the approach does not perform as well as full finetuning.

**Strengths And Weaknesses:**

The paper is well written, clear and easy to follow. The topic is quite relevant as training large models is becoming common, and finding the right ways to use the pretrained models is quite important.

However, there are a few weaknesses that should be addressed:

(1)  Table 1 results are only on small-ish pretraining datasets of ImageNet-1k and -21k. This could be limiting, as a main benefit of the contrastive training, e.g., CLIP and ALIGN has come from the massive dataset sizes (e.g., hundreds of millions to billions). It has also been shown that large image transformer models (e.g., ViT) benefit more from large supervised pretraining (e.g., billions of samples). These observations make it hard to know if the results in table 1, Fig 4, table 5 will generalize, or if they are specific to this setting. For example, adding experiments using the CLIP pretrained model, to see how that scale of pretraining data effects the results.

(2) The paper doesn't really propose anything new. It is a useful study, however, it is limited by the few settings compared. E.g., adding more layers and BiFPN for object detection, adding more layers for segmentation/action classification. These aren't especially interesting insights. One of the questions this paper sets out to address is:
"What is the key to making the frozen setting work well when the downstream tasks are significantly different from the pretraining task?"
And I don't think the current experiments really answer that question. Unless the key is simply adding more tuneable layers. Related to that, it would be interesting to compare adding trainable layers vs. making parts of the pre-trained model trainable. E.g., instead of adding 10 new layers, only add 1, but make the last 9 of the pre-trained model trainable.

There are many other options not explored in this work, which limits the insight gained from this work.

Overall, the paper has limited originality and significance due to those weaknesses.

---

> ### Author Response · Authors · 2022-08-02
> **Response to Reviewer RULo**
>
> Thanks for the valuable comments.
>
> **Q1 "Will these results change with different pretraining data scales?" "For example, adding experiments using the CLIP pretrained model, to see how that scale of pretraining data effects the results."**
>
> CLIP is developed with ViT, and since careful modifications should be made for ViT to apply to various downstream tasks, such as object detection and semantic segmentation, we did not consider ViT for study at the beginning. As it is a great suggestion to use CLIP models, we have conducted an experiment on K400 video action recognition.
>
> As shown below, when using the CLIP-400M pretrained ViT-B/16 model with a head network containing 4x Global Blocks, the top-1 accuracy on K400 of the frozen setting is even better than the full finetuning setting (79.9 vs. 78.0), which further demonstrates the benefit of freezing the backbone, that the model can better utilize the knowledge learned in pretraining while avoiding the forgetting problem.
>
>  In addition, the performance gap between the frozen setting and full finetuning is significantly reduced by using a better pretrained model with a larger-scale dataset: -4.6 for Swin-B-1K vs. -2.1 for Swin-B-22K vs. 1.9 for ViT-B-CLIP-400M. This validates that our results in Table 1 and Table 5 could successfully generalize with larger data scales.
>
> |Head Network| Frozen acc@1 | Full Ft. acc@1 |
> |--- | --- | --- |
> Linear | 70.9 | 77.6 |
> 4x Global Blocks | 79.9 | 78.0 |
>
> **Q2 "Are these methods really the key to using frozen image models? Are there other methods that could be better?" "Related to that, it would be interesting to compare adding trainable layers vs. making parts of the pretrained model trainable. E.g., instead of adding 10 new layers, only add 1, but make the last 9 of the pretrained model trainable."**
>
> We would like to first refer the reviewer to **Technical Novelty** of the general response. As discussed in the general response, this paper aims to provide a simple and feasible solution to the problem of using frozen pretrained models.  This problem is important but less-studied before. The simple solution of adding tunable and well-placed parameters is intuitive but not trivial. We hope it would be a good start for studies on frozen models.
>
> Moreover, we conduct experiments of partial finetuning with controlled trainable parameters, which has been mentioned by the reviewer.  Specifically, we experiment with three settings: finetuning the last block, finetuning the last layer, and finetuing the last layer and the last nine blocks of the penultimate layer, resulting in 33.3M / 45.9M / 76.5M tunable parameters.  As shown below, compared to partial finetuning, our solution is more efficient and effective, with fewer trainable parameters and better performance.
>
> | method | #param | Box mAP | Mask mAP |
> |  --- | --- | --- |--- |
> | Frozen backbone | 22.9M | 51.7 |45.5|
> | Finetune last block | 33.3M | 49.0 | 44.0 |
> | Finetune last layer | 45.9M | 49.3 | 44.2 |
> | Finetune last layer and last nine blocks of penultimate layer | 76.5M | 50.3 | 45.1 |
> | Full ft.| 109.2M | 51.9 |45.7|
>
> **Q3 "If this paper is trying to find the key to using frozen models, why are the results lower than finetuned? Is this an inherent limitation of using frozen models? Or has the method to properly utilize pretrained models not been found?"**
>
> Philosophically speaking, the frozen model has unique advantages and disadvantages.
>
> For the advantages:
> - It can avoid catastrophic forgetting, that is, completely retaining the information obtained from pretraining.
> - It can also alleviate over-fitting. As shown in Table 1 in the appendix, the frozen setting surpasses full finetuning on COCO when using 1\% and 10\% data.
> - It can also reduce the computation and memory overhead of full finetuning (as shown in the general response), making it affordable for low-resource institutions to use large models.
> - It can also reduce the overhead of serving models as we just need to serve a shared backbone to solve various vision problems.
>
> For the disadvantages, the frozen setting is highly demanding on pretraining. If the pretraining procedure does not cover the main information of the downstream tasks, there may be a performance loss compared to finetuning, e.g., SUP-1K (-4.6) and SUP-22K (-2.1) on K400 video action recognition as shown in Table 5. But if the pretrained model is sufficient enough, then the performance gap of the frozen and full finetuning setting can be greatly reduced, e.g., CLIP-400M (+1.9) on K400, and SUP-22K (+0.2) on COCO object detection.

---

> > ### Author Response · Authors · 2022-08-10
> > **Response to Reviewer RULo**
> >
> > Thank you again for your valuable and thoughtful comments. Do you have more comments or questions about this submission and our response? We can prepare answers or more results for them.

---

### Official Review · Reviewer_ynJ6 · 2022-07-12

**Rating:** 6
**Confidence:** 4
**Soundness:** 3 good
**Presentation:** 3 good
**Contribution:** 3 good

**Summary:**

The paper presents a comprehensive analysis of the use of pretrained image models for some major vision tasks. The paper shows the importance of well-placed tunable parameters to bridge the gap between frozen and finetuning settings, and presents an analysis of feature activation to further showcase this.

**Questions:**

See weakness.


**Limitations:**

Not adequately. There's fairness risks around using frozen pretrained models depending on the datasets the frozen models are trained on.

**Strengths And Weaknesses:**

Strength
* The paper presents an insightful analysis of the properties of frozen pretrained models for downstream tasks
* On detection tasks, the gap between frozen model and finetuning is negligible, while the gap is still there for semantic segmentation and video action recognition.
* The authors also analyze different sizes of base network and different pre-training strategy and report their properties on various downstream tasks. It’s interesting to larger models require less tunable parameter and achieve better performance without increasing trainable parameters.

Weakness:
* The results in Table 5 suggest that for some tasks frozen models are good enough (e.g. detection), but for others fine tuning still provides clear benefits (e.g. semantic segmentation, action recognition). It would be interesting to delve deeper into what causes the gap, e.g.  pretraining methods, model design, task itself, or specific dataset. For example, it’s interesting that the AP mask on COCO shows no gap between frozen vs full ft, but there’s a 2 point gap on ADE20K mIoU. I would expect the trend between AP mask and mIoU to be similar as both involve pixelwise localization. For video action recognition, I’m wondering if using a frozen model pretrained on videos (rather than images) would help bridge the gap with full ft.
* It might be helpful to show the benefits of frozen models by comparing training time or resources used.
* In Table 1, it may be interesting to add MAE (masked autoencoder) style self-supervised learning and see if that shows any difference between frozen and full ft.

---

> ### Author Response · Authors · 2022-08-02
> **Response to Reviewer ynJ6**
>
> Thanks for the valuable comments.
>
> **Q1 What causes the gap on ADE20K semantic segmentation and K400 video action recognition?**
>
> For ADE20K semantic segmentation, we agree that it's somewhat strange that the trend of ADE20K is inconsistent with COCO. To verify whether this is related to the model architecture, we have also conducted a similar comparison using the UPerNet framework, as shown in Table 2 and Table 3 in the appendix. Surprisingly, the gap between the frozen setting and the full finetuning setting almost disappears. The reason behind this phenomenon may be that the pixel decoders and Transformer decoders used in Mask2Former are hard to optimize when they get deeper (models in both the frozen and the full finetuning setting will crash when the depth of pixel decoder is larger than 6), while the optimization issue doesn't exist in UPerNet.
>
> For K400 video action recognition, the gap between the frozen and full finetuning settings is due to the pretraining task and dataset, and this can be significantly reduced by using a better pretrained model with a larger-scale dataset: -4.6 for Swin-B-1K vs. -2.1 for Swin-B-22K vs. 1.9 for ViT-B-CLIP-400M. Specifically, as shown below, when using the officially released *CLIP* pretrained ViT-B/16 model with a head network of 4x Global Blocks, the top-1 accuracy on K400 of the frozen setting is further improved over supervised pretraining with ImageNet-1K and 22K. Note that this frozen setting works even better than the full finetuning setting (79.9 vs. 78.0), which also demonstrates the benefit of freezing the backbone, that the model can better utilize the knowledge learned in pretraining while avoiding the forgetting and over-fitting issues.
>
> | Head Network   | Frozen acc@1 | Full Ft. acc@1 |
> |  --- | --- | --- |
> Linear | 70.9 | 77.6 |
> 4x Global Blocks | 79.9 | 78.0 |
>
> **Q2 "For video action recognition, I'm wondering if using a frozen model pretrained on videos (rather than images) would help bridge the gap with full ft."**
>
> We adopted VideoMAE pretrained models to explore the effect of pretraining on videos. As VideoMAE used global attention in the backbone while the main results in our work used spatial attention, we report results with both types of attention. As shown in the following table, there is still a performance gap of -4.7 (73.7 vs. 78.4) between the frozen setting and the full finetuning. This may be caused by the pretraining task of masked video modeling, which has difficulty capturing high-level semantics.
> Although the VideoMAE pretraining does not bridge the gap, the results of CLIP models in **Q1** validate that the frozen setting with a strong pretrained model would even surpass the full finetuning.
>
> | Backbone Attention | Head Network   | Frozen acc@1 | Full Ft. acc@1 |
> |  --- | --- | --- | --- |
> Spatial | Linear |20.8 | 73.3 |
> Spatial | 4x Global Blocks | 69.7 | 76.8 |
> Global | Linear | 27.6 | 76.2 |
> Global | 4x Global Blocks | 73.7 | 78.4 |
>
> **Q3 It might be helpful to show the benefits of frozen models by comparing training time or resources used.**
>
> Thanks for the great suggestion which helps us to improve this paper. We would like to refer the reviewer to the **Speed/Memory analysis** part of our general response. In the general response, we show that training with a frozen backbone can significantly improve speed and reduce memory cost and thus enabling more users with limited resources to take advantage of large models.
>
> **Q4 "In Table 1, it may be interesting to add MAE (masked autoencoder) style self-supervised learning and see if that shows any difference between frozen and full ft."**
>
> As MAE is originally developed with ViT, we compare it with other pretrained ViT models in the **Analysis on ViTs** of our general response. It is observed that MAE has low accuracy in the frozen setting, and this is very similar to SimMIM. This result may not be surprising, as MAE and SimMIM share the masked image modeling paradigm.

---

> > ### Comment · Reviewer_ynJ6 · 2022-08-08
> > **Response to author feedback**
> >
> > The authors have addressed my questions well by showing the experiments on ADE20K, K400, MAE pretraining, and conducting studies on training time/resource. It's interesting to see in Q1 the gap between frozen and finetuned models can be bridged by pretraining on  a larger dataset. The studies in Table 2 and 3 of appendix also show that the decoder head matters to the gap between frozen/finetuned models. I'd recommend a score of 'accept' for this paper.

---

> > > ### Author Response · Authors · 2022-08-09
> > > **Response to Reviewer ynJ6**
> > >
> > > We sincerely thank the reviewer for the constructive comments, which help us improve the paper. We would add the results and discussions accordingly in our revision.

---

### Author Response · Authors · 2022-08-02
**General Response to All Reviewers**

First, we sincerely thank the reviewers for their constructive feedback.

**Q1 Analysis on ViTs**

Since it is not very easy to apply ViT to various downstream tasks, such as object detection and semantic segmentation, we did not consider ViT for study at the beginning. But adding results of ViT is a good idea, so we conducted experiments with ViT-B on K400 video action recognition. Here, we test with five pretrained ViT-B models, including supervised pretraining on ImageNet-1K (SUP-1K), supervised pretraining on ImageNet-22K (SUP-22K), masked image modeling of MAE on ImageNet-1K (MAE-1K), masked video modeling of VideoMAE on Kinetics-400 (VideoMAE-K400) and vision-language pretraining of CLIP on 400M <image, text> pairs (CLIP-400M).

Following the setting of Table 1 in the paper, we used a spatial-only transformer backbone and a linear head for K400 video action recognition. As shown in the following table, MAE and VideoMAE both perform poorly in the frozen setting, which is similar to SimMIM. This result could be foreseen as they share the masked image modeling paradigm. We can also observe that the CLIP model performs best and the SUP-22K model outperforms SUP-1K by a large margin, which indicates the substantial benefit that data scaling brings to the frozen setting.

| Approach | Frozen (linear) | Full ft (linear) |
|  --- | --- | --- |
SUP-1K | 58.8 | 75.6|
SUP-22K | 64.0 | 77.1 |
MAE-1K | 28.3 | 73.9 |
VideoMAE-K400 | 20.8 | 73.3 |
CLIP-400M | 70.9 | 77.6 |

**Q2 Speed/Memory analysis**

We carefully analyzed the training speed and memory usage of **the frozen backbone with additional tunable parameters** and **full finetuning** on COCO object detection, ADE20K semantic segmentation and K400 video action recognition using various sized models from SwinV2-T to SwinV2-G. As shown below, even with more trainable parameters in the head network, training with a frozen backbone can significantly improve speed and reduce memory cost, especially for large-scale models. Moreover, freezing the backbone reduces the memory consumption of a billion-level model to less than 32G, which makes it possible to run on regular GPUs, thus helping institutions with limited resources to take advantage of such large models.

| Model | Batch Size | COCO Frozen (5x BiFPN) | COCO Full Ft. (FPN) |
|  --- | --- | --- | --- |
SwinV2-T | 16 | 0.44s / 9.74G | 0.46s / 9.78G |
SwinV2-B | 16 | 0.50s / 10.38G | 0.65s / 17.29G |
SwinV2-L | 16 | 0.57s / 10.56G | 0.84s / 25.81G |
SwinV2-G | 8 | 1.2s / 31.05G | - / > 80G |


| Model | Batch Size | ADE Frozen (6x Pixel Decoders) | ADE Full Ft. (1x Pixel Decoders) |
|  --- | --- | --- | --- |
SwinV2-T | 16 | 0.28s / 4.24G | 0.34s / 5.04G |
SwinV2-B | 16 | 0.31s / 4.42G | 0.41s / 7.84G |
SwinV2-L | 16 | 0.34s / 4.84G | 0.48s / 11.77G |
SwinV2-G | 8 | 1.23s / 23.84G| 3.08s / 78.77G |


| Model | Batch Size | K400 (4x Global Blocks) | K400 Full Ft. (Linear) |
|  --- | --- | --- | --- |
SwinV2-T | 64 | 0.31s / 10.43G | 0.51s / 15.87G |
SwinV2-B | 64 | 0.52s / 13.46G | 0.96s / 31.47G |
SwinV2-L | 32 | 0.45s / 12.80G | 0.78s / 27.05G |
SwinV2-G | 16 |1.14s / 30.54G | - / > 80G |


**Q3 Technical Novelty**

The main purpose of this paper is to design a feasible solution for frozen models on diverse computer vision tasks. For this less-studied problem, we conducted a careful empirical analysis on several key questions including what pretraining task fits best with this frozen setting, how to make the frozen setting more flexible to various downstream tasks, and the effect of larger model sizes.

Please note that adding more parameters when freezing the backbone is intuitive but not trivial. For example, for object detection, adding more parameters in FPN helps to bridge the gap, but adding in the head network does not. We provide further intuition that both the number of tunable and well-placed parameters is the key to making frozen settings work well.

Although there may be other ways to handle this problem, our study provides a strong start with a simple, practical and feasible solution. With this work, we hope to bring greater attention to this promising direction of freezing pretrained image models.

---

### Meta-Review · Area_Chair_YWAs · 2022-08-20

**Recommendation:** Accept
**Confidence:** Less certain

**Metareview:**

This paper presents a study of how well pre-trained and frozen large models work across several downstream computer vision tasks. The paper initially received mixed reviews with two of them being borderline accept and one borderline reject. The reviewers shared their concerns about the novelty of the investigation and its impact with some additional questions about the setup. The authors provided a rebuttal that addressed some of the reviewers' concerns. Two out of three reviewers updated their reviews in the post-rebuttal phase. Reviewers generally agree that the paper should be accepted but still have concerns regarding the novelty. Due to the comprehensive empirical analysis, AC recommends acceptance but suggests that the authors are urged to look at reviewers' feedback and incorporate their comments into the camera-ready.

**Award:**

No

---

### Decision · Program_Chairs · 2022-09-14

Accept